# Effects of Photoperiod Regime on Meat Quality, Oxidative Stability, and Metabolites of Postmortem Broiler Fillet (*M. Pectoralis major*) Muscles

**DOI:** 10.3390/foods9020215

**Published:** 2020-02-19

**Authors:** Jacob R. Tuell, Jun-Young Park, Weichao Wang, Bruce Cooper, Tiago Sobreira, Heng-Wei Cheng, Yuan H. Brad Kim

**Affiliations:** 1Meat Science and Muscle Biology Laboratory, Department of Animal Sciences, Purdue University, West Lafayette, IN 47907, USA; tuell@purdue.edu (J.R.T.); park1126@purdue.edu (J.-Y.P.); 2Division of Applied Life Sciences (BK 21 Plus), Gyeongsang National University, Gyeongsangnam-do 52828, Korea; 3Department of Animal Sciences, Purdue University, West Lafayette, IN 47907, USA; wang2077@purdue.edu; 4Bindley Bioscience Center, Purdue University, West Lafayette, IN 47907, USA; brcooper@purdue.edu (B.C.); sobreira@purdue.edu (T.S.); 5Livestock Behavior Research Unit, USDA-Agricultural Research Service, West Lafayette, IN 47907, USA; Heng-Wei.Cheng@ARS.USDA.GOV

**Keywords:** antioxidative status, aromatic amino acids, broiler, lighting program, meat quality, metabolite profiling

## Abstract

The objective of this study was to evaluate the effects of photoperiod on meat quality, oxidative stability, and metabolites of broiler fillet (*M. Pectoralis major*) muscles. A total of 432 broilers was split among 4 photoperiod treatments [hours light(L):dark(D)]: 20L:4D, 18L:6D, 16L:8D, and 12L:12D. At 42 days, a total of 48 broilers (12 broilers/treatment) was randomly selected and harvested. At 1 day postmortem, fillet muscles were dissected and displayed for 7 days. No considerable impacts of photoperiods on general carcass and meat quality attributes, such as carcass weight, yield, pH, water-holding capacity, and shear force, were found (*p* > 0.05). However, color and oxidative stability were influenced by the photoperiod, where muscles from 20L:4D appeared lighter and more discolored, coupled with higher lipid oxidation (*p* < 0.05) and protein denaturation (*p* = 0.058) compared to 12L:12D. The UPLC–MS metabolomics identified that 20 metabolites were different between the 20L:4D and 12L:12D groups, and 15 were tentatively identified. In general, lower aromatic amino acids/dipeptides, and higher oxidized glutathione and guanine/methylated guanosine were observed in 20L:4D. These results suggest that a photoperiod would result in no considerable impact on initial meat quality, but extended photoperiods might negatively impact oxidative stability through an alteration of the muscle metabolites.

## 1. Introduction

Broiler chicken production plays a key role in supplying consumers with high quality protein, as evidenced by the steady increase in US broiler chicken production from 50.4 billion pounds in 2008 [1] to 56.8 billion pounds in 2018 [2]. While genetic improvement plays the most critical factor in meeting this continuously rising demand [3,4], environmental factors such as rearing temperature, nutrition, and lighting schedule/intensity must be managed to fully capitalize on the genetic potential of modern broiler chickens [5,6,7].

The poultry industry has traditionally reared broilers under long photoperiods to maximize growth performance [7,8]. Several studies have identified continuous or near-continuous photoperiod regimes as positively impacting breast meat yield, feed consumption and conversion, and growth rates [9,10,11]. However, there is growing evidence that long photoperiods might negatively impact broiler health and welfare. For example, rapid growth rates in broiler chickens have been implicated in causing skeletal deformities, metabolic disorders, and increased mortality [7,12]. The increased breast meat yield commonly observed in long photoperiod regimes has often been shown to be inversely related to the yield of the thigh and drum [10,11,13], contributing to an increased frequency of leg abnormalities with impaired walking ability [9,14,15]. These detriments to skeletal health associated with long photoperiods might also associate with the disruption of the normal diurnal rhythm, which plays a critical role in regulating bone modeling/remodeling [16].

Several studies have investigated the impacts of photoperiod on broiler performance and carcass traits, but few studies have examined its potential impacts on meat quality. Previously, Li et al. [17] found that breast meat from broilers reared under a 12L:12D photoperiod has lower malondialdehyde (MDA, a secondary lipid oxidation product) concentrations compared to 23L:1D controls. Guob et al. [18] corroborated this finding, reporting higher blood serum MDA levels in broilers reared at a longer photoperiod, indicative of increased oxidative stress. It has been well-established that chronic stress is detrimental to broiler meat quality and oxidative stability [19,20,21]. However, to our knowledge, no studies have evaluated the impact of photoperiod on meat quality and oxidative stability of broiler fillet (*M. Pectoralis major*) muscles during aerobic display. Aerobic packaging (using oxygen permeable polyvinylchloride film/overwrap with a polystyrene foam tray) is the most common method of packaging for fresh broiler meat products in the US [22], despite often exhibiting a shorter shelf-life [23,24]. As such, aerobic display storage could potentially exacerbate any oxidative defect already present in fresh broiler products. The aim of this study was to evaluate the effects of photoperiod on meat quality, oxidative stability, and metabolites of postmortem broiler fillet (*M. Pectoralis major*) muscles.

## 2. Materials and Methods

All animal use and procedures were approved by the Purdue Animal Care and Use Committee (1712001657).

### 2.1. Photoperiod Treatments

Ross 308 broiler chicks (*n* = 432) at 1 day of age were weighed in groups (*n* = 18/group) and allocated among 24 pens (110 cm × 110 cm) for equal distribution of weight across the pens. The pens were randomly assigned to one of four photoperiod treatment rooms at the Poultry Unit of the Animal Sciences Research and Education Center at Purdue University. Lighting schedule regimens were performed as follows: [hours light(L):dark(D)] 20L:4D, 18L:6D, 16L:8D, and 12L:12D. For all treatments, the birds were provided with constant (24L:0D) lighting at 30 lux at 1 day of age, reduced to 23L:1D from day 2 to day 7. After this, the photoperiods were adjusted in gradual increments until reaching the final expected photo schedule at day 14, which were maintained until 42 days of age. Brooder temperature was 34 °C until day 3, after which temperature was gradually reduced until 21–24 °C was reached and maintained until 42 days of age.

All broilers were provided a starter diet with 23.43% crude protein (CP) and 3,050 kcal metabolizable energy (ME)/kg from day 1 to day 14, a grower diet with 22.81% CP and 3,150 kcal ME/kg from day 15 to day 28, and a finisher diet with 19.17% CP and 3,200 kcal ME/kg from day 29 until day 42. Food and water was provided ad libitum throughout the course of the study.

### 2.2. Harvest and Sample Preparation

At day 42, 2 broilers per pen (*n* = 12/treatment; a total of 48 broilers) were randomly selected, transported approximately 30 min, and harvested under the standard procedures. The hot carcass weight (HCW) was recorded as the weight of the carcass following plucking, evisceration, and removal of the head and feet, prior to carcass chilling. The carcasses were chilled in a commercial air cooler with an ambient temperature of 2 °C. The chilled carcass weight (CCW) was recorded as the weight of the carcass after 24 h of chilling. Cooler shrink was calculated as the percent difference between the HCW and CCW. Broiler fillet (*M. Pectoralis major*) muscles, with *M. Pectoralis minor* and skin removed, were dissected from both sides of each carcass at 1 day postmortem and weighed. Fillet yield was calculated as the weight of the fillet muscles as a percentage of CCW.

Approximately 30 g of each left side fillet muscle was collected for drip loss measurement. The remaining left side muscles were frozen and stored at −80 °C as 1 day postmortem samples until later analyses. The right side fillets were displayed for 7 days at 2 °C under 1450 lux fluorescent lighting to mimic the retail store conditions. The samples were displayed bone-side up on polystyrene foam trays with soaking pads, overwrapped with a commercial polyvinyl chloride film. After aerobic display storage, the samples were frozen and stored at −80 °C as 7 day displayed samples. Prior to analysis, the samples were frozen in liquid nitrogen and pulverized to form a homogenous powder by using a commercial blender (Waring Products, Inc., Stamford, CT, USA).

### 2.3. pH Measurement

The pH of the fillet muscles was measured at 1 d postmortem, following carcass chilling. Values were obtained from the left fillet muscles in duplicates by using a pH probe (Hanna Instrument, Inc., Warner, NH, USA) calibrated with pH 4 and 7 buffers prior to the analysis.

### 2.4. Water-Holding Capacity (WHC)

Prior to any measure of WHC, the samples were gently blotted with a paper towel to remove excess moisture from the muscle surface. Drip loss was measured in accordance with the method published by Honikel [25], with some modifications. At 1 day postmortem, approximately 30 grams of the caudal portion of the left side fillet muscles without skin and visible connective tissue was suspended with netting in an airtight container, at 2 °C for 48 h. The drip loss was expressed as the percentage difference between the weight of the sample prior to and after hanging storage. Display weight loss was determined as the percent difference between the initial and final weights of the samples before and after 7 days of aerobic display storage. Freezing/thawing loss was assessed as the percent difference between weights prior to and after freezing/thawing of 1 day postmortem samples at –80 °C and 24 h of thawing at 2 °C. Cooking loss was measured by cooking samples in a water-impermeable plastic bag submerged in an 80 °C water bath. Cooking temperature was monitored using a thermocouple (Type-T, Omega Engineering, Stamford, CT, USA) connected to a data logger (Madge Tech, Inc., Warner, NH, USA). After 71 °C was reached in the geometrical center of the fillet, the samples were immediately submerged in an ice water bath to halt the cooking process. Cooking loss was expressed as the percent change between the initial and final weight of the samples.

### 2.5. Instrumental Tenderness

The samples used for cooking loss were chilled at 4 °C for 16 h prior to the measurement of shear force. Six slices (1 cm × 1 cm) per sample were taken in a direction parallel to that of the muscle fibers. Slices were sheared perpendicularly to the fiber direction using a Warner-Bratzler type V-shaped blade attached to a TA-XT Plus Texture Analyser (Stable Micro System Ltd., UK) at 2 mm/sec. Peak shear force from the cores in Newtons was determined, and the mean values of the replicates were used for statistical analysis.

### 2.6. Proximate Composition

Proximate composition of the fillet muscles at 1 day postmortem was analyzed according to the AOAC methods [26]. Moisture was determined in triplicates at 100 °C using the oven air-drying method. Percent nitrogen was measured in duplicates following the Dumas combustion method (Leco, St. Joseph, MI, USA), and concentration of crude protein was determined by multiplying percent nitrogen by 6.25. Crude ash was measured in duplicates by combusting the dried samples in a 580 °C muffle furnace. Crude lipid was determined as follows, by the formula:100% − [% moisture + % crude protein (wet matter basis) + % crude ash (wet matter basis)](1)

### 2.7. Instrumental Color Attributes

Instrumental color was assessed daily on 7 day displayed samples. Commission internationale de l’éclairage (CIE) L*, a*, and b* values were obtained in triplicates from three randomly selected locations per fillet using a CR-400 Chroma Meter (Konica Minolta, Chiyoda, Tokyo, Japan) equipped with a CIE standard illuminant D_65_. CIE a* and b* data were used to determine the hue angle (discoloration) and chroma (saturation) values based on the American Meat Science Association meat color measurement guidelines [27].

### 2.8. Oxidative Stability and Transmission Value

Lipid oxidation was assessed using the 2-thiobarbituric acid reactive substances (TBARS) assay to determine the formation of MDA according to the method published by Buege and Aust [28] with modifications for broiler meat by Kim et al. [29]. Values of both day 1 and day 7 samples were obtained in duplicates by measuring absorbance of the obtained supernatant at 531 nm using a microplate spectrophotometer (Epoch, Biotek Instruments, Inc., Winooski, VT, USA) and multiplying by 5.54 to calculate the TBARS value. The values were expressed as milligrams MDA per kilogram of fillet muscle.

Protein oxidation, by measuring loss of thiol groups on day 1 and day 7 of aerobic display storage, was assessed in duplicates following the method published by Berardo et al. [30]. Absorbance of the sample filtrate was measured at 412 nm using a microplate spectrophotometer (Epoch, Biotek Instruments, Inc., Winooski, VT, USA), and the values of the thiol content presented in nanomoles thiol groups per milligram protein were determined using the Lambert–Beer formula (ε_412_ = 14,000 M^−1^ cm^−1^). Protein concentration of the sample filtrate was assessed using a bovine serum albumin standard curve.

Transmission values, a determinant of protein denaturation, of day 1 postmortem samples were assessed in duplicates using the method of Ockerman and Cahill [31] with modifications as described by Kim et al. [32]. Briefly, turbidity of the mixed sample, 1 milliliter of sample filtrate mixed with 5 milliliters of 0.1 M citric acid in 0.2 M sodium phosphate buffer (pH 4.6), was measured at 600 nm wavelength using a spectrophotometer (UV-1600PC, VWR International, LLC, Radnor, PA, USA).

### 2.9. Fatty Acid Profile

Intramuscular lipids were extracted from the fillet muscles in duplicates using the method of Folch et al. [33] with modifications by Shin and Ajuwon [34]. Briefly, fatty acid methyl esters (FAME) were prepared by a trans-esterification reaction, after which FAMEs were extracted in hexane. FAMEs were analyzed on a gas chromatograph (Varian CP 3900 with CP-8400 autosampler, Agilent, Santa Clara, CA, USA) equipped with a 105 m Rtx-2330 fused silica capillary GC column (10729, Restek, Bellefonte, PA, USA). Fatty acids were identified by comparison of retention times with the known standards (Supelco 37 components FAME Mix, Sigma Aldrich, St. Louis, MO, USA). Detected fatty acids were expressed in grams per 100 grams of intramuscular lipid.

### 2.10. UPLC–MS Metabolite Profiling

For metabolomics data, one sample per pen of the two extreme treatments (20L:4D and 12L:12D) was randomly selected and analyzed. Sample extraction and removal of protein was performed in accordance to the method published by Bligh and Dyer [35]. In brief, 100 mg of fillet muscle tissue powder was vortexed with 300 µL of chloroform and 300 µL of methanol, after which 300 µL of distilled water was added and centrifuged for 10 min at 16,000× *g*. The upper phase, containing the polar metabolites, was transferred to a microcentrifuge tube, evaporated to dryness with a vacuum concentrator, and reconstituted in 60 µL of HPLC diluent (95% water and 5% acetonitrile, containing 0.1% formic acid). Samples were sonicated, centrifuged, and transferred to HPLC vials.

UPLC–MS was performed using an Agilent 1290 Infinity II UPLC system (Agilent Technologies, Palo Alto, CA, USA) equipped with Waters Acquity HSS T3 separation column (2.1 mm × 100 mm, 1.8 µm) and HSS T3 guard column (2.1 mm × 5 mm, 1.8 µm) (Waters, Milford, MA, USA). A gradient of water and acetonitrile was used. Following chromatographic separation, an Agilent 6545 quadrupole time-of-flight (Q-TOF) mass spectrometer was employed. Mass data were collected and analyzed with Agilent MassHunter B.06 software from an m/z of 70–1,000. Agilent Reference Mass Correction Solution (G1969-85001) was infused to improve mass accuracy. Agilent ProFinder B.10 was employed for peak deconvolution and alignment. Peaks were annotated using the HMDB (www.hmdb.ca) metabolite database with a mass error ≤ 10 ppm.

### 2.11. Statistical Analysis

The experimental design was a randomized complete block with the photoperiod treatment (20L:4D, 18L:6D, 16L:8D, and 12L:12D) as the fixed effect and the pen (*n* = 6/treatment) as the experimental unit. Individual broilers and their interactions with the fixed effect were considered as a random effect. Data including the aerobic display storage period (i.e., color and oxidative stability) were considered as a split plot design, where the photoperiod treatment was a whole plot and the display duration was a sub-plot. Data were analyzed using the PROC MIXED and PROC GLIMMIX procedures of SAS 9.4 (SAS Institute Inc., Cary, NC, USA) with the PDIFF option for separation of the least square means (*p* < 0.05). Trends were defined as (0.10 > *p* ≥ 0.05). For metabolomics data, one sample from each photoperiod treatment lacked adequate correlation with others within the respective treatment and was omitted from the analysis, resulting in 5 replicates per treatment. Metabolomics data were analyzed using the unpaired *t*-test significance analysis, and significant metabolites (*p* < 0.05) were used for principal component analysis (PCA) modeling.

## 3. Results

### 3.1. Carcass and Meat Quality

Effects of photoperiod on carcass traits including HCW, CCW, cooler shrink, fillet weight, and fillet yield are presented in Table 1. Overall, the photoperiod did not result in any considerable impacts on general carcass quality attributes (*p* > 0.05). However, there was a tendency for carcasses from 20L:4D to lose more weight, as shown by the greater cooler shrink percentage compared to other shorter photoperiod treatments (*p* = 0.070).

Similarly, no significant effects of photoperiod on pH, WHC, and instrumental tenderness of broiler fillet muscles were found (Table 2, *p* > 0.05). There was a trend of fillets from 16L:8D showing higher freezing/thawing loss compared to others (*p* = 0.098).

Proximate composition of the fillet muscles was unaffected by the treatments as well (Table 3, *p* > 0.05).

### 3.2. Color, Oxidative Stability, and Fatty Acid Profile

There were significant interactions between the photoperiod treatment and aerobic display storage on CIE L* (lightness), CIE a* (redness), CIE b* (yellowness), and hue angle (discoloration) values of the fillet muscles (Table 4, *p* < 0.05). At day 1 of display, the muscles from 20L:4D exhibited a lighter color than the muscles from 18L:6D and 12L:12D (*p* < 0.05) but not 16L:8D (*p* > 0.05). At day 2 of display and forward, 18L:6D fillets had lower CIE L* values compared to the 20L:4D only (*p* < 0.05), while 16L:8D and 12L:12D were intermediates (*p* > 0.05). For CIE a* values, the fillets from 16L:8D were higher than other treatments on day 1 of display (*p* < 0.05), although both 18L:6D and 12L:12D were redder in color than 20L:4D (*p* < 0.05). By day 7 of display, 16L:8D fillets maintained higher redness than 20L:4D (*p* < 0.05), but were not different from 18L:6D and 12L:12D groups (*p* > 0.05). For CIE b*, there were no differences across treatments from day 1 to day 5 of display (*p* > 0.05). However, on day 6 and day 7, the fillets from 18L:6D were less yellow in color than 16L:8D (*p* < 0.05) only.

A significant interaction between the photoperiod treatment and the display time on hue angle values was found (Table 4). The highest hue angle values (indication of discoloration) were observed in 20L:4D compared to other photoperiod treatments on day 1 and day 2 of display (*p* < 0.05). This could be attributed to the lower CIE a* values observed in 20L:4D during the same display duration, coupled with numerically higher CIE b* values. From day 3 and onwards, the differences across treatments on each respective display day were less pronounced, although 18L:6D maintained a lower hue angle than 20L:4D (*p* < 0.05) but was not different compared to 16L:8D and 12L:12D treatments at any point of the display (*p* > 0.05). Chroma values were affected by display storage duration only (*p* < 0.05, data not shown), regardless of the photoperiod group (*p* > 0.05). Overall, chroma values exhibited a similar pattern as CIE b* values, increasing from day 1 to day 4 of display before decreasing by day 7.

Significant main effects of the photoperiod and the display period were observed for the TBARS values (Table 5, *p* < 0.05). Higher concentration of MDA was found in the fillet muscle samples from 20L:4D (0.50 mg MDA/kg fillet) and 18L:6D (0.52 mg MDA/kg fillet), compared to the 12L:12D (0.37 mg MDA/kg fillet) (*p* < 0.05), while 16L:8D (0.42 mg MDA/kg fillet) was intermediate (*p* > 0.05). As expected, MDA accumulated in the displayed samples from day 1 to day 7 (*p* < 0.05). There was no significant interaction between the photoperiod and the display observed in the TBARS values. There was, however, an interaction between the photoperiod and the display period for protein oxidation, as assessed by the content of the thiol groups (*p* < 0.05). A detectable loss in thiol groups was found in 20L:4D from day 1 to day 7 only (*p* < 0.05).

In terms of protein denaturation, there was a strong trend that the fillet muscle samples from 20L:4D and 16L:8D had a higher transmission than the samples from 18L:6D and 12L:12D groups (*p* = 0.058), indicating a higher degree of denaturation in the sarcoplasmic protein in those samples.

Most detected fatty acids were not different across the photoperiod treatments (Table 6, *p* > 0.05). Higher polyunsaturated fatty acids (PUFA) were found in lower photoperiod groups (16L:8D and 12L:12D) compared to 20L:4D (*p* < 0.05). Of the PUFA, higher omega-3 fatty acids were found in 12L:12D compared to the 20L:4D (*p* < 0.05), while higher omega-6 fatty acids were found in both 16L:8D and 12L:12D compared to 20L:4D (*p* < 0.05). The differences in omega-3 and omega-6 fatty acid contents were not pronounced enough to cause a significant difference in the omega-6:omega-3 ratio (*p* = 0.114), nor the ratio of unsaturated to saturated fatty acids (*p* = 0.588). There was, however, a tendency for higher palmitic (C16:0) acid in 20L:4D compared to other photoperiod groups (*p* = 0.082).

### 3.3. Metabolite Profiling

Metabolite profiling was conducted for the samples from the two extreme treatments (20L:4D and 12L:12D) in order to obtain insight into the biological and biochemical processes that might be differently affected by the photoperiod treatments. Untargeted metabolite profiling detected 1,472 metabolites in the fillet muscle samples from the 20L:4D and 12L:12D treatments. PCA was performed to discriminate between the 20L:4D and 12L:12D treatments based on the 20 significant metabolites with *p* < 0.05 (Figure 1). PC1 was shown to explain 31.65% of the total variance and PC2 as 17.02%, leading to a total PC of 48.67%.

Of the 20 metabolites found to be significantly impacted by the photoperiod treatment, 15 were tentatively identified using the HMDB metabolite database with a mass error of ≤ 10 ppm (Table 7). Overall, the muscles from 12L:12D were found to be higher in amino acids/dipeptides of aromatic amino acids (tyrosine, tryptophan, phenylalanine) with leucine/isoleucine, as well as piperidine. Samples from 20L:4D were found to have a higher oxidized glutathione, methylated histidine, and guanine and methylated/demethylated guanosine (*p* < 0.05).

## 4. Discussion

Previous studies evaluating the influence of photoperiod on broiler meat production have primarily focused on growth performance and yield [7,8]. Several studies have reported a positive impact of the length of photoperiods on growth performance during the starter phase and breast meat yield [10,11,13]. The present study observed no impact of photoperiod on HCW and CCW, agreeing with previous literatures which showed that compensatory gain occurring in the finishing stage minimizes the differences in market weight [11,13,17]. In addition, the present study found no impact of the photoperiod on the weight and yield of the fillet muscles, in disagreement to previous findings [10,11,13,17], except that the percentage of cooler shrink was linearly receded with a shorter photoperiod. In particular, there was a tendency (*p* = 0.070) for 20L:4D carcasses to lose more moisture during the carcass chilling process compared to the 16L:8D and 12L:12D groups. The difference in the current study might be caused by multiple factors, such as the rearing conditions (pen size, group density, and room temperature) or combination of test factors (light intensity, feed energy, and nutrient density). These findings do indicate, however, that the photoperiod-associated differences in meat quality cannot be attributed to weight differences of the fillet muscles.

In addition, aside from a trend of higher freezing/thawing loss in 16L:8D (*p* = 0.098), there was no other impact of photoperiod on WHC. Similarly, other studies have found no impact of photoperiod on WHC [17,36]. No differences in shear force values were also observed across treatments, and, to our knowledge, studies measuring instrumental tenderness of broiler fillet muscles associated with photoperiod effects are unavailable in the current literature. Li et al. [17] reported a higher percentage meat protein in broiler fillet muscles from 12L:12D compared to 20L:4D and 23L:1D groups. Proximate composition including crude protein concentration was unaffected in the present study. This could be explained in part by the lack of a 23L:1D group in the present study, as the meat protein values reported by Li et al. [17] were 0.67% different between the 23L:1D and 12L:12D groups but only 0.32% different between the 12L:12D and 20L:4D groups, The current and previous findings support the postulation that the photoperiod regimes of the present study are unlikely to have any considerable impacts on the composition and general meat quality attributes of the broiler fillet muscles.

In terms of oxidative stability during chilling storage/display times, photoperiod treatments had multiple main effects and two-way interactions on instrumental color and oxidative stability attributes. The fillet muscles from the 20L:4D and 18L:6D groups had higher MDA contents than the 12L:12D group. This finding is interesting considering the lower PUFA content in 20L:4D compared to 12L:12D, as well as 16L:8D, as it has been well-established that PUFA is much more susceptible to lipid oxidation [37]. However, it has been demonstrated that broiler thigh muscle is less susceptible to lipid oxidation during refrigerated storage than breast meat, despite higher free long-chain PUFA content [38]. Numerous factors can promote lipid oxidation in fresh broiler muscles including microbial growth, enzymatic activity, exposure to light and oxygen, and others [39]. As all broilers were treated in the same manner during and following harvest, it is likely differences in oxidative stability can be attributed to pre-harvest factors. The samples from 20L:4D showed a significant decrease in thiol contents from day 1 to day 7, while the samples from other groups had no difference in the thiol contents during display. The finding of longer photoperiod being pro-oxidative to broiler muscles has been corroborated by Li et al. [17] who observed a higher MDA in the fillet muscles from 23L:1D compared to 12L:12D. In addition, Guob et al. [18] reported that a 12L:12D treatment decreased the serum MDA content compared to 20L:4D. The findings were further supported by a reported trend of greater activity of superoxide dismutase, a well-known indicator of antioxidative capacity, in 16L:8D and 12L:12D treatments compared to 23L:1D and 20L:4D treatments [18]. Although antioxidant indices were not assessed in this study, current and previous data do support a positive impact of shorter photoperiod on oxidative stability. The photoperiod effect on oxidative reactions and antioxidative capacity of broiler meat would warrant further research.

The ratio of oxidized (GSSG) to reduced (GSH) glutathione, known as the glutathione redox ratio, has been used as a reliable marker of oxidative stress, based on the role of GSH in protecting against free radicals [40,41]. In the present study, UPLC–MS metabolomics tentatively identified GSSG as higher in the fillet muscles of 20L:4D compared to 12L:12D. Previously, Guob et al. [18] assessed activity of serum GSH peroxidase and found no significant relationship between its activity and photoperiod. However, Asensi et al. [40] reported a positive relationship between the glutathione redox ratio and buildup of lactate/pyruvate through exhaustive physical exercise. Thus, it is reasonable to postulate that postmortem muscle metabolism might have been altered between the 20L:4D and 12L:12D groups. The problem of pale, soft, exudative (PSE) meat in broilers has been well-documented, with findings of higher muscle CIE L*, poorer WHC, and poorer protein functionality [42,43]. The PSE condition arises from rapid glycolysis during early postmortem, causing rapid muscle acidification when the carcass is not chilled, resulting in a denaturation of muscle proteins. Given the higher CIE L* in 20L:4D early in display coupled with trends of higher moisture loss during carcass chilling (*p* = 0.070) and a great transmission value (*p* = 0.058), it would be reasonable to postulate that longer photoperiods might contribute to the PSE-like condition in broiler fillet muscles. However, given the lack of differences in pH, other measures of WHC, and metabolites related to glycolytic pathways, the hypothesis would need to be tested in further studies.

In this study, several tentatively identified metabolites with important biological functions, including guanine, methylated guanosine, and dimethylguanosine, were upregulated in 20L:4D fillet muscles compared to 12L:12D. It has been implicated that hypermethylation of purine bases is a biological marker of disrupting tumor-suppressor genes and inactivating DNA repair genes [44]. In fact, Asensi et al. [40] identified a positive relationship between glutathione redox ratio and oxidative damage of DNA. Adding to this, the current results indicated that there was an upregulation of methylhistidine in 20L:4D. A positive relationship between skeletal muscle mass [45] and turnover of myofibrillar protein [46,47] with methylhistidine has been demonstrated in humans. As broilers grow, both an increase in absolute rates of breast muscle protein synthesis and degradation is observed, leading to an overall net increase in protein deposition [48]. Thus, there is some evidence to suggest 20L:4D treatment might alter muscle metabolome in a way that would support the rapid deposition of fillet muscle tissue.

l-Phenylalanine, tryptophan-leucine/isoleucine dipeptide, and tyrosine-leucine/isoleucine dipeptide were tentatively identified and found to be upregulated in the fillet muscles from 12L:12D broilers. l-Phenylalanine and l-tyrosine are known to be the precursors for catecholamine neurotransmitters, including dopamine, epinephrine, and norepinephrine [49]. Particularly, l-tyrosine has a positive impact on reducing levels of stress hormones [50] and ameliorates negative effects of sleep deprivation [51,52], and l-tryptophan has been well-established as a precursor for melatonin and serotonin [53,54]. Melatonin has a key role in regulation of circadian rhythm, and its production is suppressed by light exposure [55,56]. Given this relationship of these amino acids to stress and diurnal rhythm, the present study provides some evidence for a mechanism of extreme lengths of photoperiods on increasing oxidative stress.

## 5. Conclusions

Photoperiod treatments had minimal impacts on the carcass and meat quality traits of broiler fillet muscles. However, color and oxidative stability were influenced by the current photoperiod treatments and aerobic display storage. In general, the 20L:4D treatment fillets appeared to be lighter and more discolored, distinct from other photoperiod treatments during early display. This was coupled with a higher lipid oxidation in 20L:4D and 18L:6D treatments compared to 12L:12D. Metabolomic analyses indicated that compared to 12L:12D, the 20L:4D group exhibited a downregulation of aromatic amino acids, known to be related to neurotransmitter production, and an upregulation of oxidized glutathione, a biomarker of oxidative stress. These findings support a potential mechanism for the generation of long photoperiod-associated oxidative defects. For practical implications, the results of this study could provide valuable information and practical insights for the poultry industry to develop some pre- and post-harvest strategies for minimizing any quality defects of fresh meat products from broilers exposed to extended photoperiod environments.

## Figures and Tables

**Figure 1 foods-09-00215-f001:**
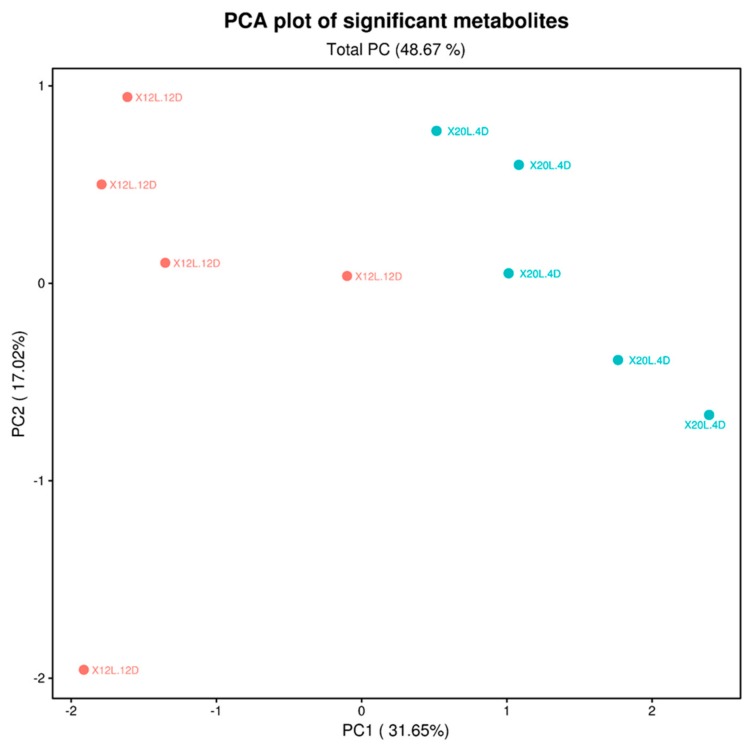
PCA modeling based on the 20 metabolites present within the broiler fillet (*M. Pectoralis major*) muscles were found to be different (*p* < 0.05) between the 20L:4D and 12L:12D photoperiod treatment groups (*n* = 5/treatment).

**Table 1 foods-09-00215-t001:** Effect of photoperiod on broiler carcass characteristics (*n* = 6/treatment).

Trait	20L:4D	18L:6D	16L:8D	12L:12D	SEM	Significance of *p*-Value
Hot carcass weight (kg)	2.28	2.42	2.22	2.38	0.07	0.199
Chilled carcass weight (kg)	2.18	2.33	2.16	2.30	0.06	0.182
Cooler shrink (%)	4.7	3.7	3.2	3.3	0.4	0.070
Fillet weight (g)	547.7	575.0	537.8	593.4	21.4	0.269
Fillet yield (%)	25.2	24.8	25.0	25.8	0.6	0.681

**Table 2 foods-09-00215-t002:** Effect of photoperiod on pH, water-holding capacity, and instrumental shear force of broiler fillet (*M. Pectoralis major*) muscles (*n* = 6/treatment).

Trait		20L:4D	18L:6D	16L:8D	12L:12D	SEM	Significance of *p*-Value
pH (24 h)		5.93	5.93	5.91	5.95	0.03	0.797
Water-holding capacity (%)	Drip loss	2.9	3.7	4.5	3.3	0.8	0.534
	Freezing/thawing loss	2.9	3.1	4.5	2.9	0.5	0.098
	Display weight loss	2.9	2.7	3.1	3.2	0.3	0.773
	Cooking loss	11.5	12.2	12.3	12.4	0.7	0.808
Shear force (*N*)		22.3	17.4	21.6	24.8	2.4	0.228

**Table 3 foods-09-00215-t003:** Effect of photoperiod on proximate composition (wet-matter basis) of broiler fillet (*M. Pectoralis major*) muscles (*n* = 6/treatment).

Trait	20L:4D	18L:6D	16L:8D	12L:12D	SEM	Significance of *p*-Value
Moisture (%)	74.6	75.0	74.4	74.2	0.3	0.350
Protein (%)	22.0	21.9	22.3	22.1	0.3	0.914
Lipid (%)	1.9	1.5	1.6	2.1	0.2	0.133
Ash (%)	1.6	1.6	1.7	1.7	0.1	0.190

**Table 4 foods-09-00215-t004:** Effect of photoperiod on D_65_ instrumental color attributes [CIE L* (lightness), CIE a* (redness), CIE b* (yellowness), hue angle (discoloration), and chroma (color intensity)] of broiler fillet (*M. Pectoralis major*) muscles during 7 days of aerobic display (*n* = 6/treatment).

Trait		*P* ^1^					Significance of *p*-Value		
	*D* ^2^	20L:4D	18L:6D	16L:8D	12L:12D	SEM	*P*	*D*	*P* × *D*
CIE L*	1 d	49.8 ^abc^	46.5 ^ij^	48.6 ^bcdefg^	47.7 ^fghij^	0.6	0.037	<0.001	0.035
	2 d	50.8 ^a^	48.8 ^bcdef^	49.7 ^abcd^	49.5 ^abcde^				
	3 d	50.8 ^a^	48.8 ^bcdef^	50.0 ^ab^	49.5 ^abcde^				
	4 d	49.6 ^abcde^	47.2 ^ghij^	49.0 ^bcdef^	48.6 ^bcdefg^				
	5 d	49.2 ^bcdef^	46.8 ^ij^	48.9 ^bcdef^	48.2 ^cdefghi^				
	6 d	49.1 ^bcdef^	46.5 ^j^	48.5 ^cdefgh^	48.1 ^efghij^				
	7 d	49.3 ^bcdef^	46.9 ^hij^	49.1 ^bcdef^	48.1 ^defghij^				
CIE a*	1 d	2.9 ^efghi^	3.6 ^bcd^	4.4 ^a^	3.7 ^bc^	0.2	0.018	<0.001	0.010
	2 d	3.0 ^efgh^	3.4 ^cde^	4.0 ^b^	3.5 ^bcde^				
	3 d	3.0 ^efgi^	3.4 ^cdef^	3.6 ^bcd^	3.4 ^cdef^				
	4 d	2.7 ^ghijk^	3.2 ^cdefg^	3.5 ^bcde^	3.1 ^defg^				
	5 d	2.4 ^ijk^	2.9 ^fghi^	3.0 ^efgh^	3.0 ^efghi^				
	6 d	2.3 ^jk^	2.7 ^ghijk^	3.2 ^cdefg^	2.5 ^hijk^				
	7 d	2.3 ^k^	2.8 ^ghijk^	2.8 ^fghij^	2.5 ^hijk^				
CIE b*	1 d	7.0 ^abcdefgh^	6.2 ^h^	7.0 ^abcdefgh^	6.5 ^efgh^	0.4	0.347	<0.001	0.026
	2 d	6.9 ^abcdefgh^	6.3 ^gh^	6.8 ^defgh^	6.2 ^h^				
	3 d	6.8 ^cdefgh^	6.2 ^h^	6.9 ^bcdefgh^	6.4 ^gh^				
	4 d	7.7 ^abcd^	7.0 ^abcdefgh^	7.9 ^a^	7.3 ^abcdefg^				
	5 d	7.5 ^abcde^	6.9 ^bcdefgh^	7.8 ^abc^	7.4 ^abcdef^				
	6 d	7.5 ^abcdef^	6.8 ^defgh^	7.9 ^ab^	7.2 ^abcdefgh^				
	7 d	7.0 ^abcdefgh^	6.4 ^fgh^	7.6 ^abcd^	7.2 ^abcdefgh^				
Hue angle	1 d	67.0 ^cdef^	59.5 ^ij^	57.4 ^j^	60.7 ^hij^	1.6	0.049	<0.001	0.007
	2 d	66.5 ^cdefg^	61.1 ^hij^	59.0 ^ij^	60.9 ^hij^				
	3 d	66.4 ^defg^	61.2 ^hij^	62.1 ^ghi^	62.5 ^ghi^				
	4 d	70.6 ^abc^	64.9 ^fgh^	65.8 ^efg^	67.2 ^bcdef^				
	5 d	71.5 ^ab^	66.5 ^cdefg^	68.4 ^abcdef^	68.1 ^abcdef^				
	6 d	72.4 ^a^	67.6 ^bcdef^	67.8 ^bcdef^	70.3 ^abcde^				
	7 d	71.7 ^ab^	66.4 ^cdefg^	69.3 ^abcdef^	70.4 ^abcd^				
Chroma	1 d	7.6	7.3	8.4	7.5	0.3	0.309	<0.001	0.208
	2 d	7.6	7.2	7.9	7.3				
	3 d	7.5	7.1	7.8	7.3				
	4 d	8.2	7.8	8.7	8.0				
	5 d	7.9	7.6	8.4	8.0				
	6 d	7.8	7.4	8.5	7.7				
	7 d	7.4	7.1	8.1	7.7				

^1^ Photoperiod effect. ^2^ Display period effect. ^a–k^ Means lacking a common superscript within a color attribute differ due to the interaction of photoperiod treatment and display period (*p* < 0.05).

**Table 5 foods-09-00215-t005:** Effect of photoperiod on the 2-thiobarbituric acid reactive substance values, thiol content, and transmission value of broiler fillet (*M. Pectoralis major*) muscles, during aerobic display storage (*n* = 6/treatment).

Trait		*P* ^1^					*D* ^2^			Significance of *p*-Value		
	*D*	20L:4D	18L:6D	16L:8D	12L:12D	SEM	1d	7d	SEM	*P*	*D*	*P × D*
TBARS ^3^	-	0.50 ^a^	0.52 ^a^	0.42 ^ab^	0.37 ^b^	0.04	0.38 ^y^	0.53 ^x^	0.02	0.038	<0.001	0.312
Transmission value (%)	-	53.2	37.1	48.8	33.9	5.4	-	-	-	0.058	-	-
Thiol content ^4^	1d	28.9 ^A^	27.0 ^AB^	27.4 ^AB^	26.8 ^AB^	1.0	27.5	26.8	0.5	0.970	0.114	0.025
	7d	25.9 ^B^	27.6 ^AB^	27.0 ^AB^	26.8 ^AB^							

^1^ Photoperiod effect. ^2^ Display period effect. ^3^ 2-thiobarbituric acid reactive substances values expressed as milligrams MDA per kilogram fillet muscle. ^4^ Thiol content expressed as nanomoles thiol groups per milligrams protein. ^a,b^ Means lacking a common superscript within a row differ due to photoperiod treatment (*p* < 0.05). ^x,y^ Means lacking a common superscript within a row differ due to display period (*p* < 0.05). ^A,B^ Means lacking a common superscript differ due to the interaction of the photoperiod treatment and the display period (*p* < 0.05).

**Table 6 foods-09-00215-t006:** Effect of photoperiod on the fatty acid profile (grams per 100 grams of intramuscular lipid) of broiler fillet (*M. Pectoralis major*) muscles (*n* = 6/treatment).

Fatty acid	20L:4D	18L:6D	16L:8D	12L:12D	SEM	Significance of *p*-Value
C14:0	0.37	0.33	0.35	0.34	0.01	0.328
C14:1	0.09	0.08	0.07	0.06	0.01	0.108
C16:0	19.9	19.1	19.2	18.9	0.3	0.082
C16:1	2.91	2.61	2.36	2.62	0.17	0.181
C18:0	7.58	7.93	7.73	7.72	0.30	0.872
C18:1(n9)	24.0	23.3	22.6	22.8	0.7	0.474
C18:2(n6)	25.6	26.5	27.2	27.1	0.6	0.234
C18:3(n3)	2.17	2.28	2.37	2.35	0.11	0.577
C18:3(n6)	0.32	0.32	0.31	0.31	0.01	0.961
C20:0	0.06	0.04	0.05	0.05	0.01	0.496
C20:1(n9)	0.19	0.19	0.18	0.17	0.01	0.776
C20:3(n3)	0.09	0.09	0.08	0.12	0.02	0.503
C20:3(n6)	1.12	1.23	1.35	1.17	0.27	0.935
C20:4(n6)	3.68	4.05	3.98	4.05	0.32	0.823
C20:5(n3)	0.26	0.25	0.23	0.25	0.02	0.873
C22:1(n9)	0.02	0.05	0.03	0.01	0.02	0.504
C22:6(n3)	0.52	0.59	0.57	0.67	0.06	0.359
SFA ^1^	27.9	27.5	27.4	27.2	0.3	0.463
MUFA ^2^	27.2	26.3	25.3	25.6	0.8	0.346
PUFA ^3^	33.7 ^b^	35.3 ^ab^	36.1 ^a^	36.1^a^	0.6	0.032
Total UFA ^4^	61.0	61.6	61.4	61.7	0.8	0.917
n3	3.04 ^b^	3.21 ^ab^	3.24 ^ab^	3.38 ^a^	0.07	0.022
n6	30.7 ^b^	32.1 ^ab^	32.9 ^a^	32.7 ^a^	0.5	0.037
n6:n3	10.1	10.0	10.2	9.7	0.1	0.114
UFA:SFA	2.18	2.25	2.25	2.28	0.05	0.588

^1^ Saturated fatty acids. ^2^ Monounsaturated fatty acids. ^3^ Polyunsaturated fatty acids. ^4^ Unsaturated fatty acids. ^a,b^ Means lacking a common superscript within a row differ due to the photoperiod treatment (*p* < 0.05).

**Table 7 foods-09-00215-t007:** Metabolites significantly different between the 20L:4D and 12L:12D photoperiod treatments (*n* = 5/treatment).

Component Name ^1^	Mass (Da)	Formula	Δppm	FC ^2^ 20L:4D/12L:12D	*p-*Value
Amino acids, peptides, and analogues					
l-Phenylalanine	165.0790	C_9_H_11_NO_2_	0	−0.8094	0.041
Tryptophan-Isoleucine/Leucine	317.1758	C_17_H_23_N_3_O_3_	6	−0.7910	0.029
N-Acryloylglycine	129.0423	C_5_H_7_NO_3_	2	−0.7386	0.036
Alanine-Isoleucine/Leucine	202.1317	C_9_H_18_N_2_O_3_	0	−0.6628	0.025
Tyrosine-Isoleucine/Leucine	294.1583	C_15_H_22_N_2_O_4_	1	−0.5159	0.043
Tyrosine-Isoleucine/Leucine	294.1579	C_15_H_22_N_2_O_4_	0	−0.3290	0.004
1-Methylhistidine/3-Methylhistidine	169.0849	C_7_H_11_N_3_O_2_	1	0.6057	0.017
Oxidized glutathione	612.1534	C_20_H_32_N_6_O_12_S_2_	2	0.8536	0.011
Nucleosides, nucleotides, and analogues					
1,7-Dimethylguanosine	311.1226	C_12_H_17_N_5_O_5_	1	0.3058	0.007
1-Methylguanosine/2-Methylguanosine	297.1091	C_11_H_15_N_5_O_5_	6	0.4195	0.009
Guanine	151.0509	C_5_H_5_N_5_O	10	0.4677	0.017
Others					
Piperidine	85.0889	C_5_H_11_N	3	−0.9575	0.041
Tyramine	137.0840	C_8_H_11_NO	0	0.2922	0.017
4’-O-Methyldelphinidin 3-O-beta-D-glucoside	479.1165	C_22_H_23_O_12_	5	0.4325	0.019
1H-Indole-3-carboxaldehyde	145.0524	C_9_H_7_NO	3	1.095	0.020

^1^ Compounds were tentatively identified using the HMDB (www.hmdb.ca) metabolite databases with a mass error ≤ 10 ppm. ^2^ Fold change from 20L:4D to 12L:12D. Positive fold change indicates higher levels in the 20L:4D group and lower levels in the 12L:12D group. Negative fold change indicates lower levels in the 20L:4D group and higher levels in the 12L:12D group.

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
