# Peer review of "Effects of Photoperiod Regime on Meat Quality, Oxidative Stability, and Metabolites of Postmortem Broiler Fillet (M. Pectoralis major) Muscles"

_foods, 2020, doi:10.3390/foods9020215_

Round 1

Reviewer 1 Report

The topic of the manuscript has a high scientific merit. The study examined the effect of different lighting schedule regimes on the growth of broilers, quality of their meat and its oxidative stability during 7 days of display.

The introduction properly supports the goal of the research. The methodology I correct and properly described. The results are properly explained and discussed. 

From my point of view, the manuscript is very well written and very interesting. It requires only a minor corrections - the only thing I would change is the tables. They need to be reorganized  - because in their current form the results are hard to compare.

Table 6. should be used as an example - it is correct.

I have used table 4 (find it in attachment) an example of how the Authors can reorganize the tables including both - the effect of photoperiod and the effect of display time. After converting this table, probably the description of the results (lines 219-238) will be easier to present. The description presented in lines 219-238 is quite confusing in its current form.

Author Response

Authors’ response:

Thanks

Reviewer 2 Report

The manuscript is carefully elaborated and clear. However, the authors should additionally add the comments below to the MS text.
What method of chilling was used: air cooling or air spray chilling
Tables 1, 2, 3, 4, 5, 6:
Number of analyzed samples in the form of n =? must be listed in the table headers
The SEM value must be reported separately for each value and group in the form (mean ± sem)
Tables 4 and 5: The name of the tables shall be supplemented by an indication whether the measurement was carried out on muscle without skin or on skin covering the muscle
Tables 4 and 5:
It seems scientifically inaccurate to say that only the effect of photoperiod was the cause of the observed differences in the parameter values monitored in Tables 4 and 5. It should be borne in mind that the observed color differences instrumentally measured in the Cielab system or TBARS values can be induced / caused by other variables / influences such as:
bleeding intensity of individual chickens, uneven localization of fat under the skin, cooling method which affects% water absorption in surface layers (REGULATION (EC) No 543/2008) and consequently water activity value, level of contamination by psychrotrophic microorganisms with proteolytic capabilities (Pseudomanedae, Proteus), intensity of their growth and reproduction, activity of lipolytic microbial enzymes
These effects should also be discussed in the relevant part of the MS text.

Author Response

Authors’ response:

Thanks
